# Structural basis for hyperpolarization-dependent opening of human HCN1 channel

Verena Burtscher[1,2,10], Jonathan Mount [2,3,4,10], Jian Huang[5], John Cowgill [1,2,6], Yongchang Chang[1,2], Kathleen Bickel[1,2], Jianhan Chen [5], Peng Yuan [2,3,4,7] ✉ & Baron Chanda [1,2,8,9] ✉

Hyperpolarization and cyclic nucleotide (HCN) activated ion channels are critical for the automaticity of action potentials in pacemaking and rhythmic electrical circuits in the human body. Unlike most voltage-gated ion channels, the HCN and related plant ion channels activate upon membrane hyperpolarization. Although functional studies have identified residues in the interface between the voltage-sensing and pore domain as crucial for inverted electromechanical coupling, the structural mechanisms for this unusual voltage-dependence remain unclear. Here, we present cryo-electron microscopy structures of human HCN1 corresponding to Closed, Open, and a putative Intermediate state. Our structures reveal that the downward motion of the gating charges past the charge transfer center is accompanied by concomitant unwinding of the inner end of the S4 and S5 helices, disrupting the tight gating interface observed in the Closed state structure. This helix-coil transition at the intracellular gating interface accompanies a concerted iris-like dilation of the pore helices and underlies the reversed voltage dependence of HCN channels.

The voltage-dependent ion channels with cyclic nucleotide-binding domains (CNBD) on their C-terminal end are a family of voltage-gated ion channels (VGICs) that exhibit unusual diversity in their voltage-dependent activity. These channels respond to depolarizing membrane voltages[1,2], but some are virtually insensitive[3–6] or respond to hyperpolarizing membrane voltages[7,8]. The hyperpolarization and cyclic nucleotide-gated (HCN) ion channels are a class of ion channels within the CNBD family that open when membrane voltage is more negative than a typical resting membrane potential. This unique property is crucial for their physiological role in pacemaking and spike synchronization[9] in the heart and nervous system[8,10,11]. HCN channels are also regulated by cyclic nucleotides, which act via the CNBD and fine-tune the frequency of their pacemaking activity. In this way, HCN

channels play a crucial role in integrating electrical signaling with a key second messenger pathway[12].

The CNBD family of ion channels shares a similar tetrameric architecture with other members of the VGIC superfamily[13]. The first four transmembrane segments constitute the voltage-sensing domain (VSD), where the fourth helix (S4) contains a repeating pattern of positive charges at every third position. These charges are the primary sensors of membrane potential. Each subunit's S5 and S6 helices come together to form the central ion-conducting pore domain. The C-linker region connects the cytosolic CNBD domain in each subunit to the S6 helix. Unlike canonical VGICs, the CNBD family exhibits a non-domain-swapped arrangement wherein the VSD of each subunit is located next to its pore helices[14–17]. These channels lack the extended S4–S5 linker

[1]Department of Anesthesiology, Washington University School of Medicine, Saint Louis, MO, USA. [2]Center for the Investigation of Membrane Excitability Diseases, Washington University School of Medicine, Saint Louis, MO, USA. [3]Department of Cell Biology and Physiology, Washington University School of Medicine, Saint Louis, MO, USA. [4]Department of Pharmacological Sciences, Icahn School of Medicine at Mount Sinai, New York, NY, USA. [5]Department of Chemistry, University of Massachusetts, Amherst, MA, USA. [6]Department of Applied Physics, KTH Royal Institute of Technology, Stockholm, Sweden. [7]Department of Neuroscience, Icahn School of Medicine at Mount Sinai, New York, NY, USA. [8]Department of Biochemistry and Molecular Biophysics, Washington University School of Medicine, Saint Louis, MO, USA. [9]Department of Neuroscience, Washington University School of Medicine, Saint Louis, MO, USA. [10]These authors contributed equally: Verena Burtscher, Jonathan Mount. ✉e-mail: peng.yuan@mssm.edu; bchanda@wustl.edu

helix that is critical for electromechanical coupling in prototypical voltage-gated ion channels. In these channels, which exhibit a domain-swapped arrangement of VSD and pore, the downward movement of the voltage-sensing S4 helix upon hyperpolarization pushes the S4–S5 linker towards the intracellular side. Thus, the S4–S5 linker acts as a lever arm that holds the pore gates closed at negative voltages by pressing on the adjacent S5 helix.

Electromechanical (EM) coupling in non-domain-swapped ion channels must involve an alternate pathway, given the absence of an S4–S5 linker helix in these channels. This noncanonical pathway of channel activation is also remarkably versatile compared to domain-swapped channels, which are all outwardly rectifying. Structures of depolarization-activated EAG channels with voltage sensors in the Down or Up configuration show that the pore is closed in both cases[14,18]. The central pore is also occluded in the structures of HCN channels where the voltage sensors are either in the resting (Up) or activated (Down) conformation[17,19]. The S4 helix, in both channels, exhibits a break in the middle in the Down conformation, suggesting that this is a distinct mechanistic feature of HCN channels[19,20]. Nevertheless, mainly due to a lack of structures corresponding to a hyperpolarization-activated Open state, the central question of how the HCN voltage sensor in the Down state drives pore opening remains to be determined. A recent structure of rabbit HCN4 with an Open pore and resting voltage sensors likely corresponds to an intermediate state rather than a bona fide open HCN channel[21].

In this study, we present the structures of the human HCN1 isoform in Open, putative Intermediate, and Closed conformations. Our structures reveal that the intracellular ends of the S4 and S5 helices progressively unwind as the channel undergoes closed to open transition upon hyperpolarization. These structural changes at the electromechanical coupling interface are accompanied by iris-like dilation of the pore gates in the Open-state structure. Our findings provide insights into the mechanism of reversed gating polarity in HCN channels.

## Results

To determine the structures of hyperpolarization-activated ion channels in various gating states, we focused our investigation on a human HCN1-F186C-S264C channel mutant, which was first described by Lee and MacKinnon[19]. Metal cross-bridging of the two introduced cysteines locks the voltage-sensing domain in an activated conformation as the S4 helix is shifted down about one helical turn compared to its position in the resting state. Remarkably, the S4 helix in this structure is interrupted by a break in the middle, causing the lower part of this helix to become almost parallel to the membrane plane. Although this mutant is functionally locked open by metal cross-bridging in biological membranes, the pore helices in the corresponding cryo-EM structure are tightly shut. Thus, this structure corresponds to a putative intermediate or a pre-open state (PDB: 6UQF)[19], where the voltage sensor is activated, but the pore is closed.

### Structures of HCN1 channel in three conformations

The pre-open-state structure of HCN1 was solved in digitonin, but the structure of the rabbit HCN4 isoform solved in LMNG exhibited a different conformation[21]. In this HCN4 structure, the pore appears open despite no bound cAMP, and the voltage sensor is in the resting conformation. This likely corresponds to a rare instantaneous Open state observed at depolarized potentials. We used size-exclusion chromatography to screen various detergent-lipid conditions for solubilization to explore whether the detergents and lipids may influence the channel's conformation. We found that HCN1-F186C-S264C mutant in a mixture of 0.0025% LMNG, soy polar lipids, and cholesterol hemi-succinate elutes primarily as a monodisperse peak (Supplementary Fig. 1). This fraction was collected, and its structure was determined in the presence of $Hg^{2+}$ to a global resolution of 3.6 Å using

single-particle cryo-EM (Fig. 1a, right panel, Supplementary Figs. 2 and 3, and Supplementary Table 1). In the cryo-EM reconstruction, the cytosolic region, including the CNBD, is well-resolved with side-chain densities for most amino acids. However, while transmembrane helices S5 and S6 are well-resolved, the other four, especially S1 to S3, are at lower resolutions.

Nonetheless, the unique, bent geometries of S1-S2 are consistent with rigid-body motion from the closed-state structure. At the same time, densities for bulky side chains such as W145, F166, W175, W221, Y277, H279, and W281 facilitate an unambiguous registry (Supplementary Fig. 3). S3 appears to have undergone a minor conformational change in the Open state. The C-terminal S3b helical fragment is less bent than in the Closed structure. The activated voltage-sensor HCN structure (PDB: 6UQF)[19] was used as the initial model to build the atomic model of the open-state HCN1 channel.

To ascertain the role of the detergent-lipid mixture as opposed to a metal cross-bridge in stabilizing the open conformation, we also solved the structure of this mutant under reducing conditions without changing the detergent-lipid mixture to a resolution of 3.16 Å (Fig. 1a, left panel, Supplementary Figs. 4 and 5, and Supplementary Table 1). This structure is nearly identical to the closed-state structure reported previously (PDB: 5U6P)[17]. The voltage-sensing S4 helix is in the Up position, and the central ion conduction pathway is closed. Thus, we conclude that the Open channel conformation is primarily facilitated by oxidizing conditions that trap the voltage sensors in a hyperpolarized conformation.

Upon further examination of the cryo-EM densities, we discovered that C309 in the S5 helix and C385 in S6 likely form a disulfide bridge in the Open but not in the Closed state (Supplementary Figs. 3 and 5). Both these cysteines are conserved in mammalian HCN isoforms. Substituting C309 with alanine results in a functional channel with a voltage dependence that is shifted towards more negative potentials, indicating that it is more difficult to open the mutant channels (Supplementary Fig. 6). To obtain a complete estimate of the energetic cost of disulfide cross-bridge in stabilizing the Open state, an assessment of the open channel probability (Po) is required, which is technically challenging given the low single-channel conductance of these channels[22].

To investigate the role of these cysteines further, we solved the structure of the HCN1-F186C-S264C-C309A mutant in the presence of $Hg^{2+}$ and an identical LMNG:lipid:CHS ratio to a global resolution of 3.6 Å (Fig. 1a, middle panel, Supplementary Figs. 7 and 8, and Supplementary Table 1). This cryo-EM density map is of higher overall quality than the Open structure and has fewer regions with broken density. Based on the position of the S4 and S5 helices, we classify this as the Intermediate-state structure (Fig. 1b, c). Finally, it is worth noting that in both Open and Intermediate structures, we observe only weak densities corresponding to the N-terminal HCN domain, suggesting that interactions with this domain and the S4 helix or CNBD are weakened upon activation.

To study how voltage-sensor conformations drive channel gating and minimize the confounding effects of cAMP binding, all three structures were obtained in the presence of saturating concentrations of cAMP. A comparison of the Open and Closed structures reveals that while the open channel conformation involves tilting of S2, rearrangement of S3b, and downward translocation of S1 and S4, the S1–S4 bundle itself is not radially displaced (Fig. 1d). In comparison to the apo HCN1 closed state (PDB: 5U6O)[17], we did not register a cAMP-mediated upward movement of the CNBD domain towards the transmembrane domain, unlike the reported upward movement in the bacterial cyclic nucleotide-gated ion channel[23]. We observed several lipid-like densities in the Closed state that were absent in the Open state, potentially due to the lower resolution in that region. Interestingly, we observe far fewer lipid densities in this Intermediate structure, with only 1–2 prominent densities at the S5–S6 interface than in the Closed state. Nevertheless, the superposition of the Open and Closed structures reveals that the movement of the pore helix in the Open state would sterically require the displacement

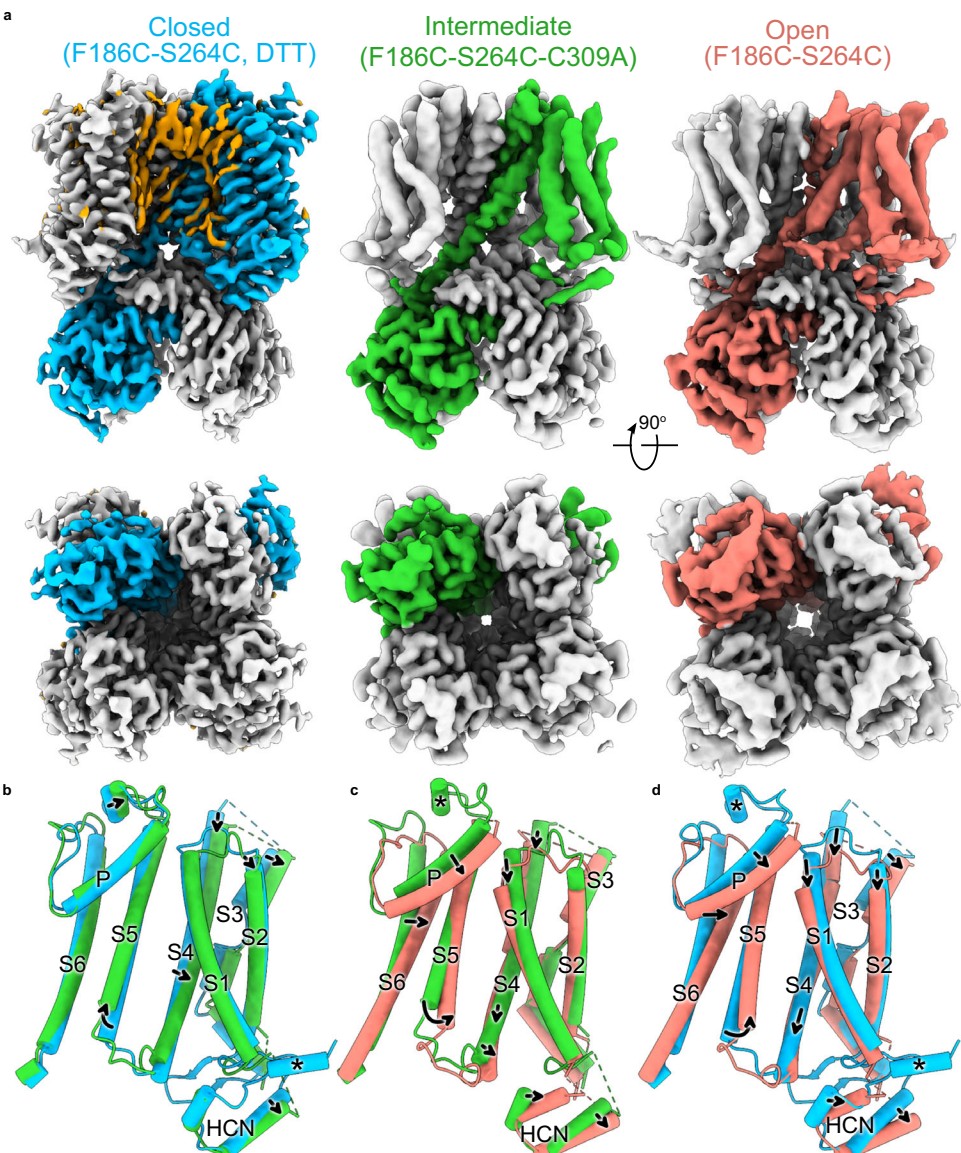

**Fig. 1 | Structures of the Closed, Intermediate, and Open conformations of the HCN1 channel. a** Cryo-EM density map corresponding to the three conformations. The top panel shows the side view and the lower panel shows the bottom view of the structures. One protomer in each of the three structures is colored either sky blue (Closed), lime green (Intermediate), or salmon (Open). Putative lipid densities were only observed in the Closed conformation and are colored orange. b-d, Side view of pairwise superposition of the various protomers: **b** the Closed and Intermediate state; **c** the Intermediate and Open state; **d** the Closed and Open state.

of the annular lipids observed in the Closed structure (Supplementary Fig. 9), indicating that opening necessitates rearrangement of the surrounding lipids.

## Voltage-sensor conformations

Although the voltage-dependent activity of HCN channels is reversed, the voltage-sensing S4 segment moves in the same direction as depolarization-activated ion channels[19]. In other words, membrane hyperpolarization drives downward movements of S4 segments in all voltage-gated ion channels. A comparison of the voltage-sensing domains in the three HCN channel structures reveals that in the Intermediate state, the R4 charge of the S4 helix moves past the Cα position of the charge transfer center F186 by a quarter helical turn (Fig. 2a). In the Open state, the R4 charge moves past this position by a complete helical turn (Fig. 2b) and the S4 helix retains its mid-helical bend in the Open state. However, unlike the pre-open structure, it does not break or become parallel to the membrane plane (Fig. 2c)[19,20]. The bend angle is similar to the Down-state structure of the EAG1 obtained

in hyperpolarized membranes (Fig. 2d; PDB: 8EP1)[18]. Another unique feature is that the intracellular end of the S4 helix unwinds and shortens by about two helical turns in the Open-state structure. While the resolution at the intracellular end of the S4 helix in the Intermediate state precludes definitive assignment of secondary structure, the weaker density at the C-terminal end suggests that it may be on the pathway to unraveling. Note that this unwinding of S4 helix is not observed in the Pre-Open voltage-sensor structure[19]. Despite the shortening of the HCN1 S4 helix in the Down state, it still extends at least two helical turns further than the EAG1 S4 helix in the Down state (Fig. 2d)[19].

## Pore conformations and ion permeation

While the central pore of the HCN channel is nonconductive in the Closed and Intermediate states, it is dilated in the Open state (Fig. 3a). Using the HOLE software, we were able to obtain a quantitative estimate of the width of the ion-conducting pore. Figure 3 shows that the Closed structure has two major constrictions: one in the selectivity

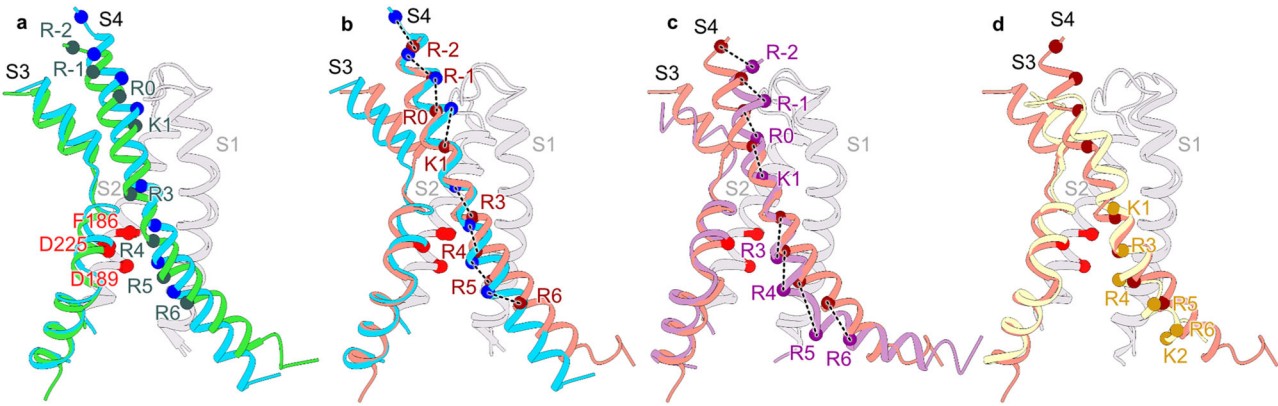

**Fig. 2 | Relative movements of the S4 helix during voltage-gating.**
**a, b** Comparison of the S4 helix position between a, the Closed and Intermediate state, and b, the Closed and Open state. The Cα atoms of positively charged residues in the S4 helix are depicted as spheres and are labeled as R-2 (R252), R-1 (R255), R0 (R258), K1 (K261), R3 (R267), R4 (R270), R5 (R273), R6 (R276). The Cα atoms of the residues F186, D225, and D189, constituting the charge transfer center, are represented by red spheres. Dashed lines indicate the displacement of each charge.

c-d, Comparison of the S4 helix position between the HCN1 Open state with either (**c**) the voltage sensor activated pore structure of HCN1 (6UQF) or **d** the closed pore EAG1 structure with the voltage sensor in the resting state (8EP1). Cα atoms of positively charged residues in the S4 helix of the open EAG1 channel are depicted as spheres and labeled as K1 (K327), R3 (R330), R4 (R333), R5 (R336), R6 (R339), and K2 (K340). All the structures were aligned with respect to the S1 and S2 helices.

filter and the other in the bottom intracellular gate. In both the Closed and Intermediate structures, the diameters of the pore gate are below 1.0 Å, too small to allow passage of ions. While the Intermediate structure shows a marginal increase in the diameter of the cavity and narrowing of the gate and selectivity filter, there is no noticeable rotation of the S6 transmembrane helices. However, the diameter of the pore gate in the Open structure increases significantly to about 7.4 Å, driven by the dilation of the S6 helices and rotation of the residues Y386, T394, and Q398 away from the pore axis. Note that the rearrangement of S6 requires concerted movement of S5 helices to prevent steric clashes. Interestingly, the P-helices rotate clockwise and fill the growing gap between the upper portion of adjacent S6 helices upon channel opening, presumably stabilizing the open-pore conformation. Comparison with the open-pore structure of HCN4 (PDB: 7NP3)[21], in particular the pore-lining S6 helices, indicates that our HCN1-F186C-S264C structure represents an open conformation (Supplementary Fig. 10a). Notably, the selectivity filter in the Open state undergoes conformational rearrangement simultaneously with gating transitions at the pore gates, leading to pore expansion in the selectivity filter (Fig. 3a and Supplementary Fig. 10b). Similarly, in the open-pore structure of HCN4, the structure of the selectivity filter differs from its closed-state structure.

To evaluate the ion permeation properties of the Open-state pore, we first examined the stability of the Open and Closed structures in all-atom molecular dynamics simulations (MD) (Supplementary Fig. 11). Both structures maintained stable root mean square deviations (RMSD) after the initial equilibration phase in their unrestrained simulations, indicating the stability of the resolved cryo-EM structures. The root mean square fluctuations (RMSF) averaged from four protein monomers are largely below 2 Å except for long loops and the terminal tails. Interestingly, the selectivity filter region in the Open state shows greater flexibility compared to that of the Closed state (Supplementary Fig. 11a, lower panel), which is likely due to the dilation of the filter (Fig. 3a). The pore profiles of both the Open and Closed structures are stable, displaying small fluctuations throughout the unrestrained simulations (Supplementary Fig. 11b, left panel). The Open state consistently maintained a larger radius in the narrow constrictions along the central pore and remained hydrated (Supplementary Fig. 11b, right panel). In contrast, in the Closed state there exists a dry region in the lower pore gate region (Z -0 to 15 Å), consistent with a nonconductive state. The ability of the Open-state pore to remain hydrated suggests that it is likely conductive to ions.

To further test if the dilated pore in the HCN1 Open-state structure is potassium permeable, we performed all-atom MD simulations with an applied voltage of −500 mV. In three parallel simulations lasting 500 ns each, we observed 5, 4, and 6 ion permeation events corresponding to a conductance of 3.2 ± 0.5 pS, which is in the ballpark of reported experimental values[22,24–26]. Figure 3b plots all five events from one of the trajectories, with representative snapshots from typical events shown in Fig. 3c (see also Supplementary Movie 1). We observe that the selectivity filter region alternates between "one-ion" and "two-ion" occupancy states during permeation events, similar to what was previously proposed for HCN4 channels[21]. The carbonyl oxygen atoms from residues I359 and C358 form two separate coordination sites for potassium ions, each of which can coordinate one potassium ion[27–29]. However, the filter region of HCN1 is more dynamic, and the switching between "one-ion" and "two-ion" occupancy states was not as regular as in HCN4. The ring of Y385 side chains form the narrowest constriction in the Open State (Fig. 3a), where the permeating ion would pause for up to ~100 ns before crossing over. This may explain the modest pS level conductance of HCN1 channels. Analysis of the potassium free-energy profile, derived from the −500 mV simulations, show only modest barriers of ~2 kcal/mol along the central pore, located near residues C358 of the filter region and Y386 in the upper pore (Supplementary Fig. 12). Both regions are where the pore opening is the narrowest (Fig. 3a). They undergo transient wetting/dewetting transitions and are only partially hydrated (Supplementary Fig. 12). The permeating ions can lose ~3 coordinating water molecules in the filter region, and ~2 coordinating waters around Y386 due to contributions from the backbone carbonyl groups or tyrosine hydroxyl groups (Supplementary Fig. 12). It was observed that the side chains of Y386 could form an inter-locked hydrogen bonding ring, with an average hydrogen bond number of 1.22 per side chain, which leads to long wait times (~100 ns) for potassium permeation. Besides, the permeating potassium ions display a significant tendency to stay near the pore central axis (within an offset distance of <2 Å) within the TM region, except at the C358 site where the ions need to avoid the dewetted environment and steric repulsion from the side chains of C358 (Supplementary Fig. 12, right panel).

## Cytoplasmic domain rearrangements
All inwardly rectifying ion channels in the VGIC superfamily comprise a conserved CNBD that follows the six transmembrane helices on the C-terminal end. While in HCN channels, the binding of cyclic

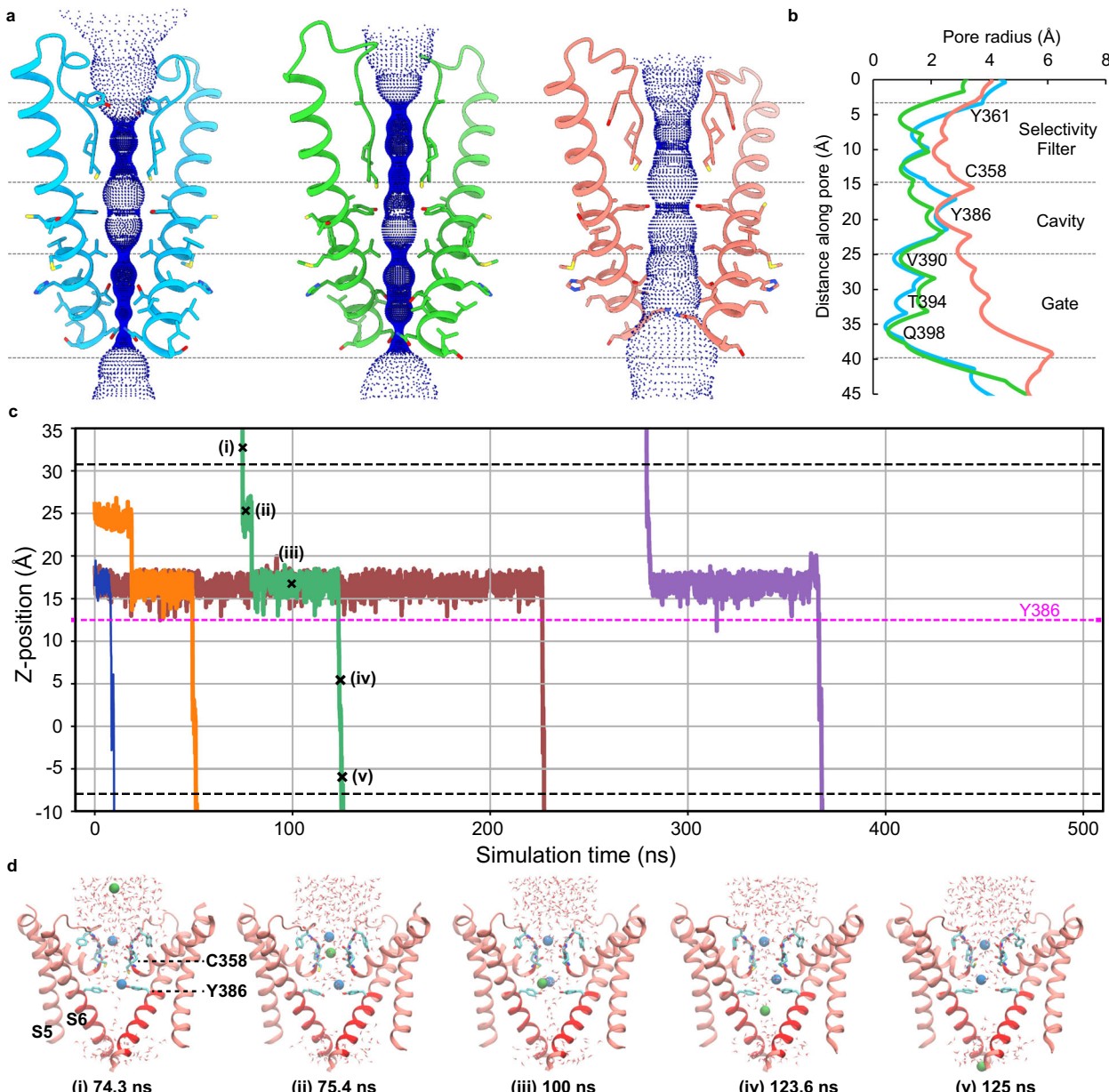

**Fig. 3 | Pore dilation in the Open state permits potassium permeation.**
**a** Comparison of the solvent-accessible pathways in the Closed, Intermediate, and Open states and their corresponding pore radii (**b**). The pore-lining S6 helices and the selectivity filter of two opposing subunits (ribbon) are depicted with the corresponding solvent-accessible pathway generated by the HOLE program. Pore-lining residues are shown as sticks. **c** Atomistic molecular dynamics (MD) simulation captures the permeation dynamics of potassium ions through the Open HCN1 channel structure under an applied hyperpolarizing potential of −500 mV. Trajectories depicting the Z-positions of permeating potassium ions along the pore axis are plotted over time. Five distinct permeation events are observed during the 500 ns simulation. The Z-position reference point is anchored to the center of mass of the bundle-crossing lower gate (residues 394–398) set at Z = 0 Å. Dashed black lines denote the presumed positions of the membrane interfaces, and the dashed magenta line denotes the position of residue Y386. **d** Representative snapshots of a potassium permeation event (corresponding to the crosses marked on the green trajectory in (**b**)). For clarity, we depict residues 297–400, which include the S5 helix, P-helix, selectivity filter, and the S6 helix from two diagonal protomers, in the cartoon representation. The permeating potassium ion is illustrated as a green sphere, whereas the positions of other potassium ions within the pore region are shown as pale blue spheres. The red region in the cartoon denotes the restrained areas during the simulation. Critical pore-lining residues in the filter, along with Y386 and water molecules within a specified cutoff radius of 12 Å around the central pore axis, are all depicted using the stick representation.

nucleotides to this domain promotes voltage-dependent opening, the role of this domain in other hyperpolarization-activated ion channels such as KAT1 channels remains unclear[30]. The effect of cAMP binding on voltage-dependent activation is isoform-dependent; for HCN4 and HCN2, voltage-dependence shifts by 15–30 mV, whereas HCN1 shifts by ~5 mV in the presence of cAMP[31–34].

The cryo-EM density maps of the cAMP-bound CNBD are generally of higher resolution than the transmembrane regions and are similar to those described in previous cryo-EM structures of HCN channels[17,19,21]. The structures of the CNBDs in the Closed, Intermediate, and Open states are similar but not identical (Fig. 4). The four cyclic nucleotide-binding domains assemble into a symmetric gating ring below the pore, which is stabilized by extensive inter- and intra-subunit interactions in the C-linker region that connects the CNBD to the S6 pore helix (Fig. 4a). Pore opening causes a slight expansion of the CTD mainly due to the rearrangement of the C-linker region. Local superposition of the

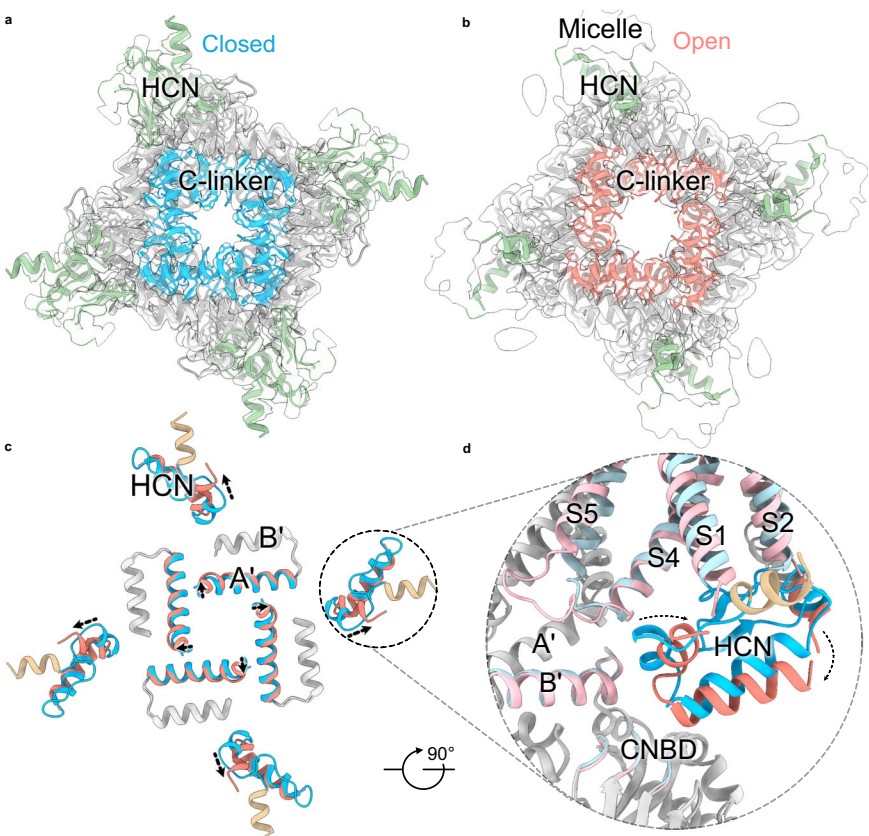

**Fig. 4 | CTD ring expansion and HCN domain movement during channel gating.**
**a**, **b** Top view of the intracellular domains in the Closed (**a**) or Open (**b**) conformations. Protein structures and corresponding density maps depict the HCN domains (pale green), CNBD (gray), and C-linker (sky blue in Closed and salmon in Open). **c** Comparison of the interface between the C-linker (A′-B′ helices) and the HCN domain (top view). The structures were globally aligned, and the HCN domain's resolved helices and the C-linker A′-helix were colored sky blue (Closed)

or salmon (Open). Arrows indicate the CTD ring's expansion and the HCN domain's rearrangement. The complete HCN domain was only resolved in the Closed state but not in the Open state where the distal N-terminal helix (tan) was not resolved. **d** Zoomed-in side view of the HCN domain-B′ helix-CNBD interface compared between the Closed and Open conformations. The color coding of the HCN domain is the same as in (**a**, **b**).

isolated C-linker and CNBD reveals a few notable structural changes in the CNBD upon pore opening (Fig. 4b). The inter- and intra-subunit interface of the C-linker region and CNBD region changes upon channel gating, the primary difference being displacement of the A′ helix in the C-linker region, and an interaction of the E′-helix with one of the strands on the β-jelly roll present in the Closed conformation, but not in the Open-state structure, where this helix is poorly resolved. In addition, there is a loss of interactions between the HCN domain and the C-linker, along with those involving the S4 helix, in the Open state. We should note that the contribution of HCN domain may differ in HCN1 channels compared to HCN4 channels, where they have been shown to be involved in channel gating.

## Electromechanical coupling

To investigate the changes in structure related to electromechanical coupling, we focused on the S4 and S5 helices at the gating interface of all three structures (Fig. 5). In the Closed state of all hyperpolarization-activated ion channels, the S4 and S5 helices are closely packed and connected by a short loop[19,21,35]. Mutations in this crucial gating interface have been shown to disrupt the ability of the channels to close[36]. Strikingly, in the Intermediate and Open-state structures, the intracellular helix–turn–helix region between I284 and V296 becomes unwound (Fig. 5a and Supplementary Fig. 13), suggesting that voltage-sensor activation destabilizes this gating interface. Concomitant with these structural changes in the S4 and S5 helices, the residues in this region make extensive contact with the B′ helix of the C-linker region in the Open state. In Closed state, in contrast, the residues in the S4–S5

loop form tight interactions with those in the A′ helix of the C-linker (Supplementary Movies 2 and 3). Although the S4 and S5 helices in the Intermediate and Open states are shorter than in the Closed state or the Pre-Open state, this decrease in length may not fully account for pore dilation in the Open-state structure.

A comparison of the two structures (viewed from the intracellular side) reveals that the primary difference is in the direction of the bend relative to the central pore axis. In the Pre-Open structure, the lower end of the S4 helix is bent towards the central pore axis, whereas in our Open structure, it points away from the pore axis (Supplementary Fig. 14). In the Closed-state structure, the full-length S4 helix is parallel to the central pore axis. Taken together, these findings suggest that the combined unwinding and bending of the HCN S4 helix, concomitant with the outward tilting of the bottom of S5 helix away from the central pore axis, drive the conformational changes that ultimately lead to channel opening (Fig. 6).

## Discussion

Most of the structures of hyperpolarization-activated ion channels published to date correspond to channels in a closed state where the voltage sensor is in the Up conformation[17,21,35]. These structures share an unusual feature that hints at the mechanism of reversed gating polarity in these ion channels. Specifically, the S4 helix in these structures is notably longer than S4 segments in depolarization-activated ion channels and protrudes to the intracellular side, where it interacts with the arm of the C-linker from a neighboring subunit[37]. This characteristic feature has been observed in both plant KAT1 and

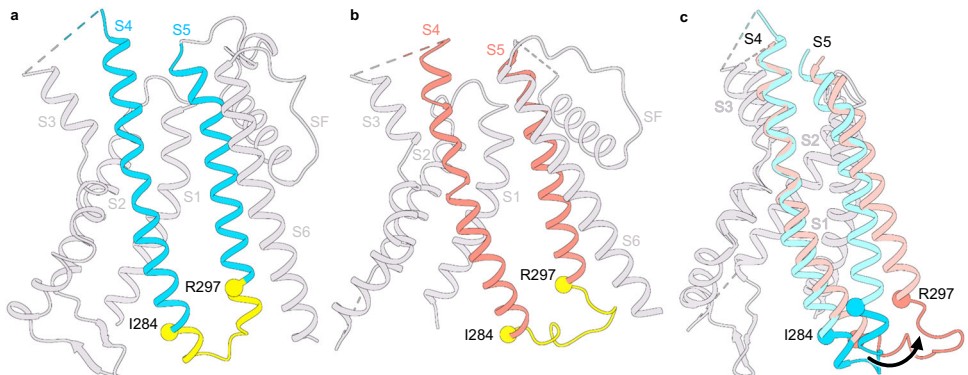

**Fig. 5 | Conformational changes at the electromechanical coupling interface.** **a**, **b** Structures of the transmembrane region corresponding to the Closed (**a**) and Open (**b**) conformations. S4 and S5 helices are either sky blue (Closed state) or salmon (Open state). Residues I284 and V296 (denoted as spheres) demarcate the intracellular gating interface (in yellow) that undergoes helix coil. **c** Superposition showing the side view of S1–S5 helices of the Closed (sky blue) and Open (salmon) states. The P-loop and the S6 transmembrane helices were removed for clarity. Structures were aligned to helices S1 and S2.

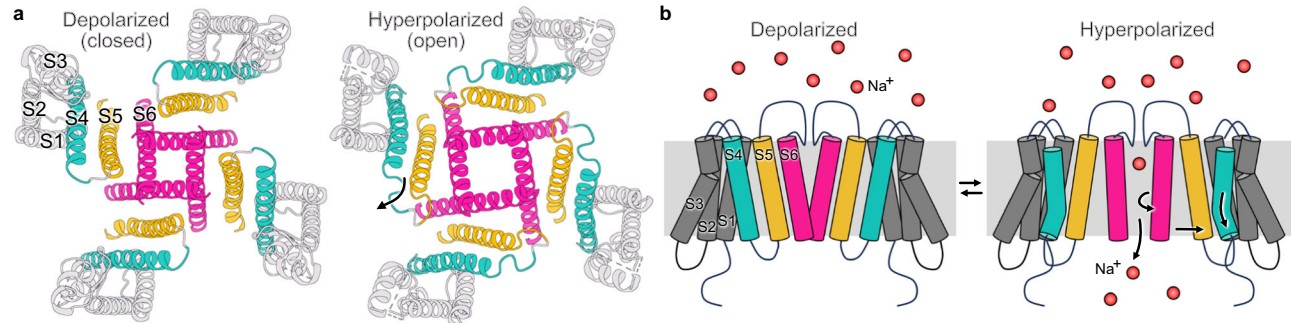

**Fig. 6 | Relative motions of the electromechanical coupling interface and pore helices during channel gating.** **a** Bottom view of S1–S6 helices of the Open (left) and the Closed (right) state. Pore loop, selectivity filter, HCN domain, and CTD were not shown for clarity. The coloring of the helices is as follows: S4 (turquoise), S5 (gold), S6 helices (magenta), and S1–S3 helices (light gray). The arrow shows the displacement of the bottom end of the S4, S5, and S6 helices in the hyperpolarized state compared to the depolarized state. **b** Schematic showing side view featuring transmembrane helices from two opposing subunits. The color coding is the same as in (**a**). The arrows highlight the displacement of the S4–S6 transmembrane helices in the Open (hyperpolarized) relative to the Closed (depolarized) state.

animal hyperpolarization-activated (HCN) ion channels[17,35]. It has led to the hypothesis that the interactions between the lower end of the long S4 helix and the C-linker region stabilize the closed pore and are critical for electromechanical coupling[17]. Although deleting the C-terminal domain, which includes the C-linker and CNBD, did not affect the voltage dependence of channel opening, its effect on the strength of electromechanical coupling has not been ascertained due to the low single-channel conductance of these channels[32]. Mutagenesis studies, including disulfide cross-linking experiments, indicate that the conformation of the C-linker and its interaction with transmembrane regions of the HCN domain is important for hyperpolarization-dependent opening[37–41]. Thus, the C-linker may be a key mediator of reversed voltage-dependent gating[37,42].

Our structures of the human HCN1 isoform uncover several features that are common in both Open and Intermediate conformations. First, the S4 helix is bent in the middle but not as much as the activated voltage sensor in the Pre-Open structure of HCN channels (PDB: 6UQF)[19]. Indeed, the bend angle is more comparable to that of the S4 helix in other channels like TAX-4 and EAG1, both belonging to the CNBD family (Supplementary Table 2)[16,18]. The bend angle is also remarkably similar to the predicted activated state structure based on FRET measurements[43]. Strikingly, the lower end of S4 helices is directed away from the central pore axis in our Open-state structure, in contrast to the pre-open state (PDB: 6UQF)[19]. In that structure, the lower ends of the S4 helices are bent towards the central pore axis, which may partly account for pore closure despite an activated voltage sensor (Supplementary Fig. 14).

Second, the intracellular ends of S4 and S5 helices undergo a helix-coil transition resulting in disruption of the tight intracellular gating interface. Mutations near this interface increase the voltage-dependent open probability, indicating destabilization of the closed state. For instance, a salt bridge between residues R297 and D401 is important for stabilizing the channel in closed but not open conformation[37,44]. Third, the interaction between the S4–S5 loop and the C-linker is altered. Instead of interacting with residues in the A' helix of the C-linker (in the Closed state), the residues in the unstructured S4–S5 region interact with residues in the B' helix. Finally, voltage-dependent pore opening is accompanied by rearrangements in the tetramerization interface mediated by the C-linker and CNBD. These findings are consistent with functional data suggesting that the C-terminal domain facilitates pore opening at hyperpolarized potentials[37–41].

Another surprising finding is the role of interaction between S5 and S6 in channel opening. Specifically, the C309–C385 disulfide bond formation is only observed in the Open state but not in the Closed state. Mutation of C309 to alanine results in an intermediate state with closed pore supporting the notion that—at least structurally—this bond is stabilizing the Open-state structure. Our functional data suggest that the disulfide bond marginally stabilizes the open pore but is not critical. The disulfide bond between C309 and C385 is a strained disulfide staple conformation that is a hallmark of allosteric disulfide bonds[45,46]. Such high-strain disulfide bonds have been found to regulate the activity of various enzymes, are highly labile, and their formation is conformation-specific[47]. Nevertheless, we cannot rule out the

possibility that the disulfide bond observed in our structure is due to our experimental conditions and does not have a physiological role.

All-atom simulations of ion permeation through the Open-state structure in a lipid bilayer with an applied voltage clearly show that this channel pore is conducting. Note that the selectivity filter of the HCN4 Open-state structure also looks different (and dilated) compared to the Closed-state structure. Selectivity filter dilation has been a subject of extensive investigation and underlies C-type inactivation in voltage-gated potassium (Kv) channels[48–50]. To the best of our knowledge, the eukaryotic HCN channels do not undergo C-type inactivation. Our study does not directly shed any light on these intriguing differences between the two types of channels, but some of them likely arise from differences in the structures of the selectivity filter and the composition of residues in this region.

By comparing the structures of HCN channels in the three different conformations, we propose a mechanism of reversed electromechanical coupling. An applied voltage moves the voltage sensor downward past the charge transfer center, causing the intracellular ends of the S4 and S5 helices to unwind in a concerted manner, leading to the disruption of the tight intracellular end of this gating interface. The loss of interactions at this interface and the bending of the lower S4 away from the central pore axis allows the bottom half of the S5 helix to tilt radially outward (Fig. 6). These structural changes facilitate the outward tilting of the S6 helix, resulting in pore dilation (Supplementary Movie 4). The diameter of the gate at residue Q398 increases to ~6 Å, which is sufficient for hydrated potassium ions to permeate through the pore. Surrounding lipid molecules will also likely play a major role in stabilizing the intracellular gating interface[21,35,51]. We can resolve lipids in the Closed-state structure but not in the Open conformation. It is unclear whether these differences are due to the lower resolution of Open and Intermediate-state structures or reflect state-dependent displacement of interacting lipids. Sea urchin HCN channels (spHCN) can be trapped in closed and open conformations by cross-linking of the S4–S5 linker with the C-linker region[38,41]. Consistent with these findings, we observe the S4–S5 linker interacts with the A′helix in the Closed state, whereas in the state, it interacts primarily with the B′ helix. The disulfide crosslinks in HCN channels may reposition the C-linker, stabilizing the preceding S6 pore helix in either conformation, depending on the free-energy landscape.

To gain insight into mechanisms that result in opposing gating polarities, we compared the recently reported resting-state structures of the EAG1 (PDB:8EP1)[18] channel with our structure of the activated, open HCN channel. The most notable difference between the two channel types is how the S4–S5 helices interact with each other in a state-dependent fashion. In EAG channels, the S4 helix is of standard length, and in Down conformation, the S4 helix is positioned close to the S5 due to the interaction between the two helices, which keeps the pore closed at negative voltages. In HCN channels, on the other hand, the extended S4 helix forms a tight interface with the S5 and stabilizes the closed pore conformation at depolarized potentials. When the membrane is hyperpolarized, the downward movement of HCN S4 disrupts the intracellular gating interface largely due to a loss of secondary structure in the lower part of the S4–S5 helices. This loss of stabilizing interactions initiates further rearrangements of the S5 and S6 helices, culminating in pore opening (Fig. 6).

While the concerted unwinding of multiple helices during channel gating is unusual, individual transmembrane helices have been shown to undergo helix-coil transitions. For example, the S6 helix of TRPM8 channels undergoes this transition during activation[52]. Similarly, the intracellular end of the S4 helix of the Nav1.7 channel also unwinds when the voltage sensors move to the Down conformation[53]. We note that the intracellular ends of the HCN S4 and S5 helices are situated outside the bilayer, particularly at hyperpolarized potentials, making this secondary structure transition energetically feasible. This transition will incur a larger energetic penalty in the lipid membrane, where there are no water molecules to form hydrogen bonds with the backbone amides. It is also possible that some of the energetic costs associated with these structural transitions result in weak allosteric coupling between the voltage sensor and pore in these channels. Nevertheless, further experimental evidence is necessary to evaluate the proposed model of HCN channel activation and determine if this mechanism applies to other hyperpolarization-activated ion channels.

## Methods
### Ethical statement
The experiments with Xenopus oocytes were performed in accordance with the approved Institutional IACUC (AWA #D16-00245) animal protocol no. 23-0156.

### Molecular biology and protein purification
The human HCN1-EM construct in the pEG vector was used as a background for all the mutants in this study[17]. The C-terminal region spanning amino acids 636–865 is deleted, and the N-terminus is tagged with eGFP, followed by an HRV 3C protease cleavage site. We added a twin-strep tag on the N-terminus for affinity purification[54]. HCN1-EM-F186C-S264C and HCN1-EM-F186C-S264C-C309A mutants were made using the FastCloning technique[55] with primers 5′-CTTCTGCTGGCCCATTGGGATGGGTGTCTTCAG-3′ and 5′-CCCAATGGGCCAGCAGAAGCATCATGCCGATC-3′ and verified by Sanger sequencing before use.

Both HCN1-EM-F186C-S264C and HCN1-EM-F186C-S264C-C309A were expressed in suspension cultures of Freestyle HEK cells using a BacMam system[56]. Bacmid DNA containing these constructs was amplified in DH10Bac competent cells (Bac-to-Bac; Invitrogen). Sf9 cells were transfected with Bacmid DNA (1–2 µg/$10^6$ cells) using Cellfectin II (Invitrogen). The supernatant containing P1 baculoviruses was harvested after 5 to 7 days, sterile filtered, and used to generate P2 baculovirus (dilution factor: 100). The P2 baculovirus was sterile filtered and supplemented with 2.5% FBS before use. For transduction, an 800 mL suspension culture of Freestyle HEK cells (3–3.5 × $10^6$ cells/mL) was transduced with 3.5–5% baculovirus. After 10–12 h at 37 °C, the cells were supplemented with 10 mM sodium butyrate and kept at 30 °C for another 48–50 h. The cells were harvested by low-speed centrifugation, and cell pellets were stored at −80 °C.

For purification, the cell pellet of 400–1200 mL suspension culture was lysed with 50–80 mL hypotonic lysis buffer (20 mM KCl, 10 mM Tris, protease inhibitor cocktail (Sigma Aldrich, P8340), pH 8.0) and sonicated 6–8 times for 10 s. The membrane fraction was collected by spinning at 50,000× $g$ for 45 min. The membrane pellets were solubilized in 70 mL solubilization buffer (300 mM KCl, 40 mM Tris, 4 mM DTT, 10 mM LMNG (NG310, Anatrace), 4 mM CHS (CH210, Anatrace), protease inhibitor cocktail (P8340, Sigma Aldrich), pH 8.0) for 1.5 h. After removing the insoluble fraction by centrifugation, the supernatant was incubated with Strep-Tactin Sepharose (iba, #2-1201-025) for 2.5 hrs, packed onto a column and then washed with 3.5 column volumes of wash buffer (300 mM KCl, 20 mM Tris, 2 mM DTT, 0.5 mM LMNG, 0.1 mM CHS, 164.5 µM SPL (541602, Avanti Polar Lipids), 10 µM cAMP, pH=8.0). The protein was eluted in 3–6 mL wash buffer containing 10 mM d-desthiobiotin. After removing Twin-Strep-tag and GFP using 3C protease, the protein was further purified using a Superose 6 Increase 10/300 GL column (Cytiva Life Sciences, # 29091596) in a running buffer (150 mM KCl, 20 mM Tris, 0.025 mM LMNG, 5.4 µM CHS, 7.5 µM SPL and 10 µM cAMP, pH=8.0). To obtain the closed-state structure, the SEC buffer was supplemented with 2 mM DTT. For all samples, 200 µM HgCl was added, and the protein samples were concentrated to 1–2.5 mg/mL using a concentrator (cutoff: 100 kDa, Amicon Ultra, Millipore). Representative SEC profiles and Coomassie blue-stained SDS-PAGE gels for the Closed, Intermediate, and Open-state structures are depicted in Supplementary Fig. 1.

## Cryo-EM grid preparation and image acquisition

Quantifoil R-1.2/1.3 or R-2/2 holey carbon copper grids were cleaned for 60 s using a Gatan Solarus 950. Before plunge freezing, the samples were spiked with 1 mM fluorinated fos-choline-8 (Anatrace, F300F) and incubated for 1 min at 100% humidity and 4 °C using a FEI Vitrobot Mark IV.

The closed-state structure was obtained from a single dataset collected on a 200 kV Glacios TEM equipped with a Falcon IV detector (ThermoFisher Scientific). Movies were collected at a magnification of ×150,000. The calculated pixel size was 0.928 Å, and the nominal defocus was set between −0.8 and −2.4 μm. A dose rate of approx. 5.43 e⁻/Å²/s was used with a cumulative dose of 53.07 electrons per Å². The intermediate and the open-state structures were obtained from four datasets collected on a 300 kV Titan Krios TEM equipped with a Falcon IV detector at a magnification of ×59,000. The calculated pixel size was 1.086 Å, and the nominal defocus was set between −0.8 and −2.4 μm. The dose rate used was in the range of 3.99–4.74 e⁻/Å²/s and did not exceed a cumulative dose of 55.07 electrons per Å².

## Cryo-electron microscopy image processing

A detailed overview of the data-processing workflow of the Closed, Intermediate, and Open-state structure is provided in Supplementary Figs. 1–3. In all three cases, movie frames were aligned and dose-weighted using patch-based motion correction and then subjected to patch-based contrast transfer function (CTF) determination in CryoS-PARC 4.4[57]. After patch-based motion correction[58] and CTF estimation[59], manual curation of the dataset was performed based on estimated CTF resolution, cross-correlation, and ice contamination. Particles were picked using a blob-based autopicker within a 150–300 Å range and subjected to 2D classification. 2D classes corresponding to intact HCN1 channels were combined and used to generate an ab initio model with C1 symmetry. Heterogeneous refinement with the initial model and bait classes was performed to sift for intact channels with high-resolution features. This class underwent C4 nonuniform refinement[60] and was utilized to generate 2D projections corresponding to multiple possible channel orientations. These projections were lowpass filtered to 20 Å to prevent Einstein-from-noise or model bias and were used for particle repicking, resulting in slightly higher particle count and improved orientational sampling. In the intermediate and open datasets, picked classes from this step were used to train Topaz on a subset of 300 micrographs. Particles corresponding to intact channels identified by either blob picking, template-based picking, or Topaz-picking were pooled, and duplicate particles were removed. A soft mask around protein density was generated for use with local refinement. While all PHENIX refinement and model building was done against global b-sharpened maps, supplementary DeepEMhance[61]-sharpened maps were also deposited.

For the Closed conformation, 1973 exposures were manually curated to remove low-quality micrographs, resulting in 1734 that were retained for further processing. Blob-based picking identified 610,860 particles, which were subjected to 2D classification to identify particles representing intact channels. These 93,472 particles were utilized to generate templates for particle repicking. Template-based picking identified 698,204 particles that were subjected to 2D classification—103,753 were retained for further processing. Particles were subsequently separated into three classes using heterogeneous refinement with C4 symmetry. Two of the classes represented intact HCN1 channels and were essentially identical, while the remaining class had broken density, was at a much lower resolution, and was discarded. The two identical classes were combined, subjected to nonuniform refinement with C4 symmetry imposed followed by local refinement within a soft mask, and underwent iterative per-particle CTF-refinement defocus correction and reference-based motion correction until map quality could no longer improved. The final 88,923 particles were refined to

3.16 Å resolution as assessed by gold-standard Fourier shell correlation (FSC).

To obtain the Intermediate conformation of HCN1, 2310 of 2575 exposures were used for downstream analysis. Blob-based particle picking, as described above, identified 936,163 particles, 126,671 of which were retained after 2D classification. After ab initio reconstruction followed by C1 heterogeneous refinement and C4 nonuniform refinement, the resulting model was used to project 2D templates for template-based picking. Template-based picking identified 1,842,624 particles, 125,994 of which were retained for downstream analysis. These particles were used to train Topaz on a subset of 300 micrographs. Topaz identified 430,812 particles. After two rounds of 2D classification, 162,648 particles corresponding to intact channels remained. Just as was performed for the blob-picked particles—particles identified by template picking or Topaz were subjected to C1 ab initio reconstruction and heterogeneous refinement to separate particles corresponding to intact HCN1 channels. The final class from each determined subset was pooled, and duplicate particles were removed. Ab initio reconstruction with a 6 Å cutoff separated 122,422 particles with high-resolution features that did not improve with further classification. The final 122,422 particles were refined with nonuniform refinement to 3.71 Å resolution as assessed by gold-standard FSC. Optimizing per-particle defocus, EWS correction, and local refinement improved the final map resolution to 3.58 Å. The final half-maps were sharpened with a global B-factor of -80.

For the HCN1 open conformation, 8196 movies were initially collected. Because analysis of this dataset revealed that the open-state particles had significant orientation bias, an additional dataset with 3947 movies spaced evenly between 20°, 30°, and 40° was collected, and these data were merged, manually curated, and 11,958 total micrographs were used for analysis. Blob picking was performed as described above, and 849,016 particles corresponding to intact channels were identified after subsequent template-based particle repicking. After two rounds of 2D classification and ab initio reconstruction followed by C1 heterogeneous refinement, the highest quality particles were used to generate a C4 symmetric map for the purposes of template generation and further particle picking. After two rounds of 2D classification, intact channel particles from template picking were used to train Topaz on a set of 200 micrographs from the initial dataset and 100 micrographs from the tilt series data. Topaz identified 2,916,674 particles, which were subjected to two rounds of 2D classification, ab initio reconstruction, and C1 heterogeneous refinement, resulting in 532,209 particles. Final particles identified from blob, template, and Topaz-picking were merged and duplicates removed, resulting in 750,743 particles for downstream processing. The particles were imported into Relion 4.0[62] and subjected to 3D classification without orientational assignment to identify channels with improved TM density. After separation into 6 classes, 3 classes with the highest resolution in the TMD were combined, and the corresponding 438,138 particles were imported into CryoSPARC for nonuniform refinement with C4 symmetry followed by local refinement within a soft mask, resulting in a 3.6 Å map. The final half-maps were sharpened with a global B-factor of −100.

## Model building and validation

For building the initial models, the cAMP-bound HCN1 closed-state structure (PDB: 5U6P)[17] or cAMP-bound HCN1 pre-open-state structure (PDB: 6UQF)[19] was rigid-body fitted into the cryo-EM density maps of the Closed or Intermediate and Open state in a segment-based manner using COOT[63,64], respectively. Unresolved regions were deleted from the model, and secondary structure features and side chains that did not fit the density were manually rebuilt and refined in COOT. To avoid overfitting, residues with missing side-chain density were trimmed. In iterative cycles, the atomic models were manually corrected for side-chain outliers in COOT and real-space refined in PHENIX[65]. The final

atomic models were validated using MolProbity[66] and the RCSB PDB validation server. Cryo-EM data collection and refinement statistics are summarized in Supplementary Table 1. Structural illustrations were prepared with UCSF ChimeraX[67].

## Molecular dynamics simulations

The cryo-EM models of the Open and Closed state of the HCN1 channel were used to build the simulation systems. The structures were embedded in the 1-palmitoyl-2oleoyl-sn-glycero-3-phosphatidylcholine (POPC) lipid bilayer and solvated using TIP3P water using the CHARMM-GUI web server[68,69]. 900 mM KCl was used to neutralize and buffer the system. The final simulation systems contain ~58,000 TIP3P water molecules, ~440 lipids, and ~1900 ions, with a total atom number of ~ 265,000 and dimension of $135 \times 135 \times 140$ Å$^3$. The protein and ions were described using the CHARMM36m force field[70], and the POPC lipids were described using the CHARMM36 lipid force field[71]. All simulations were performed using GPU-accelerated GROMACS 2019[72]. A cutoff of 12 Å and a smoothing switch function starting at 10 Å were used to calculate the non-bonded forces. The Particle Mesh Ewald (PME) algorithm was employed to calculate the electrostatic interactions[73]. All hydrogen-involved bonds were constrained using the SHAKE algorithm[74]. All simulations were performed under 298 K and maintained using the Nose–Hoover thermostat[74], with a pressure of 1 bar in the $x$–$y$ plane of the system box, using the Parrinello–Rahman barostat algorithm[75].

For equilibrating the initial systems, the systems first underwent 5000 steps of energy minimization using the steepest descent algorithm and then were subjected to a total 6 equilibration steps where the heavy atoms of protein and lipids (and ligands) were harmonically restrained with the restraining force constant gradually decreasing from 10 to 0.1 kcal mol$^{-1}$ Å$^{-2}$, as recommended by CHARMM-GUI[69] The missing loops of the protein (residue index 1–108, 130–139, 196–208, 243–251, 299–295, 325–339, 607–615, and 636–890 in the Open state and residue index 1–93, 243–251, and 636–890 in the closed state) were not fixed. Instead, we employed positional restraints of 1 kcal mol$^{-1}$ Å$^{-2}$ to the breaking sites with respect to the reference positions in the cryo-EM structures. We performed two groups of simulations under NPT conditions: the first group of simulations was three parallel 100 ns normal molecular simulations of the closed and open-state systems; the second group of simulations was also three parallel 500 ns simulations of the open-state system with applied electric field of −500 mV to mimic the hyperpolarization condition where the HCN1 channel is conductive under physiological environment. Of note, in the applied electric field simulations, the heavy atoms of the selectivity filter region (residue 358–361) and the C-α atoms of the lower half of the S6 helices (residue 386–400) were harmonically restrained using restraining constants of 0.5 kcal mol$^{-1}$ Å$^{-2}$ and 1 kcal mol$^{-1}$ Å$^{-2}$, respectively, to their reference positions of the cryo-EM structure.

All trajectories were first aligned using the transmembrane helices' backbone (residue index 140–166, 173–195, 209–242, 253–287, 297–324, and 369–401) prior to all analysis. The GROMACS rmsf and rms module were used to calculate the RMSF and RMSD profiles. The MDAnalysis python package[76,77] was used to calculate the Z-positions of permeating potassium ions, hydrogen bond analysis, estimated ion permeation PMF profiles and hydration level profiles with self-designed python scripts. All cartoon snapshot representations were plotted and rendered using the VMD software[78].

## Heterologous expression and electrophysiology

$h$HCN1-EM or $h$HCN1-EM-F186C-S264C-C309A were subcloned into a pUNIV vector[79] and cRNAs were synthesized using a mMessage T7 transcription kit (Invitrogen). Isolated *Xenopus laevis* oocytes were injected with 20–40 ng cRNA and incubated at 17 °C for 24-48 h before recording. Two-electrode voltage-clamp (TEVC) recordings

were obtained at room temperature using a CA-1B amplifier (Dagan) at a sampling rate of 10 kHz and were filtered with a cutoff frequency of 5 kHz. Electrodes were fabricated from thin-walled glass pipettes (World Precision Instruments) using a P97 micropipette puller (Sutter Instruments). The pipette resistance was in the range of 0.5–0.9 MΩ using 3 M KCl. The bath solution contained 107 mM NaCl, 5 mM KCl, 20 mM HEPES, and 2 mM MgCl$_2$ (pH = 7.4 NaOH). For channel activation, the oocytes were held at a holding potential of −20 mV and were then conditioned to potentials ranging from −20 to −110 mV with 10 mV decrements before pulsed to 0 mV at which the tail currents were recorded. The leak-corrected peak amplitude of the tail current reflects the fraction of activated channels and was plotted as a function of the conditioning pulse. The activation curve was normalized and fitted with a single Boltzmann function:

$$\frac{G}{G_{max}} = \frac{1}{1 + exp^{\frac{V - V0.5}{k}}}$$

where $V$ is the conditioning pulse, $V_{0.5}$ is the potential of half-maximal activation and $k$ is the slope factor. Data were collected and analyzed using pCLAMP10.0 (Molecular Devices) and plotted using OriginPro 2020b (OriginLab Corp., Northampton, MA). Data shown in Supplementary Fig. 6 represent mean ± SEM.

## Reporting summary

Further information on research design is available in the Nature Portfolio Reporting Summary linked to this article.

## Data availability

The cryo-EM maps have been deposited in the Electron Microscopy Data Bank (EMDB) under accession codes EMD-41036 (the closed-state conformation of hHCN1-EM-F186C-S264C); EMD-41041 (the open-state conformation of hHCN1-EM-F186C-S264C); and EMD-41040 (the intermediate-state conformation of hHCN1-EM-F186C-S264C-C309A). The atomic coordinates have been deposited in the Protein Data Bank (PDB) under accession codes 8T4M (the closed-state conformation of hHCN1-EM-F186C-S264C); 8T50 (the open-state conformation of hHCN1-EM-F186C-S264C); and 8T4Y (the intermediate-state conformation of hHCN1-EM-F186C-S264C-C309A). To assist with model building, we utilized previously reported PDB structures as initial models—they can be found under accession codes 5U6P (HCN1-EM in complex with cAMP in a depolarized closed conformation); and 6UQF (HCN1-EM-F186C-S264C in complex with cAMP in a hyperpolarized conformation). The source data underlying Figs. S1 and S6—including uncropped gels and electrophysiology data—are provided as Source Data files. The atomic coordinates for the initial and final configurations of the molecular dynamics trajectories in Fig. 4, S11, and S12 have been supplied as Source Data files. Source data are provided with this paper.

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

## Acknowledgements

The research was supported by NIH grants to B.C. (NS101723), J.C. (R35 GM144045) and P.Y. (GM143440). V.B. is supported by a Schrödinger fellowship (J 4652) by the Austrian Science Foundation. This research was partially supported by a fellowship from the University of Massachusetts as part of the Chemistry-Biology Interface Training Program (National Research Service Award T32 GM139789) (to J. H.). We thank R. MacKinnon for the generous gift of the *h*HCN1-EM pEG and the *h*HCN1-EM-F186C-S264C pEG plasmids, as well as C. Czajkowski for the pUNIV vector. We thank Drs. B.T. Summers, K. Basore, and M. Rau at Washington University Center for Cellular Imaging (WUCCI) for grid freezing, technical advice, and cryo-EM data collection.

## Author contributions

V.B., J.C. (Cowgill), and B.C. conceived and designed experiments. V.B. optimized conditions for cryo-EM experiments and performed protein purification with help from J.C. (Cowgill). J.M. performed cryo-EM data analysis and atomic model building. V.B. conducted electrophysiology experiments, Y.C. and K.B. generated the mutants, and J.H. performed molecular dynamics simulations. V.B., J.M., J.H., J.C (Chen), P.Y., and B.C. analyzed the data and wrote the manuscript with input from J.C. (Cowgill), Y.C., and K.B. All authors contributed to reviewing and revising the manuscript.

## Competing interests

The authors declare no competing interests.
