## [Peer Review File · Nature Communications]

Structural Basis for Hyperpolarization-dependent Opening of the Human HCN1 ChannelREVIEWER COMMENTS

Reviewer #1 (Remarks to the Author):

The authors here presented three different cryo-EM structures of HCN channels with closed, intermediate, and open states. They compared the different structures and found that a single S4 gating charge moves past the charge transfer center from the closed state to the intermediate state when S4 moves inward but the pore stays closed. While two gating charges move inward and past the charge transfer center from the closed state to the open state. In addition, the inward S4 movement unwinds the S4-S5 linker and breaks the interaction important for closing the channel in the cytosolic ends of S4 and S5. This would move the lower parts of S5 and S6 together to open the gate. The data seem convincing, but the conclusions seem inappropriate based on the data presented. The intermediate state of the HCN channels needs more evidence and validation. The authors need to explain more about some missing structures (for example, the lipids and HCN domain) in the intermediate and open states. The model of the voltage sensor-to-gate coupling for the reversed gating in HCN channels is not new and exciting. There are some concerns below that need to be addressed.

1. Despite the lower resolution, what are the other possibilities of undetectable HCN domains in both the intermediate and open states?
2. Could the authors explain more about the S4-S5 linker being disordered? Is there any experimental evidence? The data from Yellen's lab about Cd²⁺ crosslinking the missing S4-S5 linker with the C linker in different states seems to contradict the authors' suggestion that the S4-S5 linker is disordered. How would one explain locking the channel in the open state if there is no rigid structure to bind to?
3. Figure 4 (and in movie 6 on BioRxiv), the selectivity filter seems to expand a lot in the open state, how come the authors never mentioned it in the text? How does that change the HCN channel's selectivity?
4. The narrowest part of the selectivity filter seems to be at C358, which is different from the closed and intermediate states. Any explanations? If this C358 is important for the SF, more experimental data would be needed to validate the open structure. For example, the authors can mutate this C358 and see if it changes ion selectivity.
5. In Figure 2C and Line 161, this is a strong statement. The S4 is two helices shorter compared to a previous structure. Is it indeed shorter, or the missing two helices is due to the low-resolution structure?
6. Fig S7, Since C309 and C385 can form a disulfide bridge in the open state and HCN1-F186C-S264C-C309A produced an intermediate state of the HCN channels, have the authors tested HCN1-F186C-S264C-C385A as well, structurally and functionally?
7. Also, what is the role of the intermediate state? Any evidence for this intermediate state in other data or physiological function of this state?
8. The authors keep mentioning that the S4-S5 inner ends keep the channels closed, please elaborate more.

Reviewer #2 (Remarks to the Author):

The nonselective voltage-gated cation hyperpolarization and cyclic nucleotide gated (HCN) channels play critical roles in pace making and spike synchronization in the heart and nervous system and are implicated in epilepsy and neuropathic pain. Structures of HCN1 and 4 channels have been solved in different states, with closed or open pore but neither of these structures had voltage-sensing domains (VSDs) in the active position and the channel pore open at the same time. Burtscher et al. used the HCN1-F186C-S264C construct with VSDs locked in the active position previously developed by MacKinnon lab and protein solubilization conditions discovered by Saponaro et al. to capture three conformations of the channel in the putative closed, intermediate, and open states. Based on

comparison of these states, the authors propose the mechanism of HCN channel activation by membrane voltage.

Given common doubts about structures of putative open states representing conducting ion channels (doi: 10.7554/eLife.58660), presentation of geometrical dimensions of the pore is not a convincing argument towards an open conducting state. It would be greatly beneficial for this manuscript if authors could present molecular dynamics simulations (e.g., see ref. 21) illustrating that the conformation of HCN1, which they call open, conducts water and ions. This is especially important in this case because the resolution of the putative open-state structure is poor and the information for side chain orientations is largely missing. Needless to say, if authors can improve the resolution of their open-state structure, that would greatly strengthen the manuscript.

More specific points:

- 1) Line 102. The transmembrane helices S5 and S6 are not well-resolved. There is almost no density for side chains in these helices, which is quite poor for the nominal 3.73Å resolution.
- 2) Line 105. By looking at cryo-EM density, it is hard to conclude that the registry in S4 is unambiguous. Instead, it is rather ambiguous. The N-terminal part of S4 is asking for being moved up by 1.5-2 helical turns into unaccounted density above. The density bump assigned to R270 can easily correspond to W221 or R273. The entire C-terminal part of S4 fits very poorly into the map, emphasized by unaccounted density next to I278. With such poor overall placement, it is very hard to justify a bump of density-based assignment of W281 as well.
- 3) Line 107. S3 does not look like a single helix. It appears to be two consecutive helices tilted relative to each other. Alternatively, S3 can be called a single helix severely kinked in the middle.
- 4) Lines 116-122. When deciding between what was more important for the open state, detergent-lipid mixture or metal cross-bridge, the authors solved the structure in reducing conditions but in the same detergent-lipid mixture. However, to prove their point that the oxidizing conditions primarily facilitate the open state, the authors need to solve the structure in the presence of Hg²⁺ and DDM and show that the channel is still open. Otherwise, both factors are important contributors to their open channel conformation.
- 5) Lines 132-134. S5 and S6 are shown together in Extended Data Fig. 2, where one can appreciate the density for the putative disulfide bridge. In Extended Data Fig. 5, S5 and S6 are shown separately and there is no chance to make an opposite observation. Please include a comparable view of S5 and S6 in Extended Data Fig. 5.
- 6) Lines 161-162. If S4 moves up in the cryo-EM density (see comment 2 above), would it really be shortened?
- 7) Lines 187-188. Here, Y386, V390, and Q398 are called gate residues. Should these be V390, T394, and Q398 instead (see Figure 4)? The authors also mention the rotation of these residues. However, how do they know that these residues rotate, if the side chains are not visible in the cryo-EM density (see, for example, V390)?
- 8) Line 204. Reference 32 reports crystal, not cryo-EM structures.
- 9) Line 205. To better illustrate that the CNBD structure in all three states is identical, please add a panel with their superposition to Figure 5.
- 10) Lines 233 and 393. The PDB code does not match the reference.
- 11) Lines 250-255. Based on the resolution, model building in this region is very approximate. Accordingly, it seems to be a stretch to discuss how much the disulfide bond is strained guided by such a low-resolution model.
- 12) Figure 1 and further. The colors chosen for the closed and open conformations are very similar. It would be beneficial to change the color of one (to blue?) to make comparisons clearer.
- 13) Please label structures in Extended Data Figure 3 and/or indicate which one is which in the figure legend.

Response to Reviewers:

We are grateful to both the reviewers for their constructive feedback and critiques. This has pushed us to add new data which has significantly enhanced the quality of this manuscript. To highlight the new findings, we have made many changes throughout the manuscript as listed below:

1. The abstract was reduced to 150 words, as required by Nature Comms.
2. MD simulations showing ion permeation through the Open pore structure added to Figure 3.
3. New cryo-EM data shows densities corresponding to the S4-S5 linker which is now modeled and shown in Figure 5.
4. The main text body has been rearranged, and modifications are highlighted in grey.

Reviewer #1 (Remarks to the Author):

1. Despite the lower resolution, what are the other possibilities of undetectable HCN domains in both the intermediate and open states?

With our new cryo-EM data of the Open state structure, we can resolve parts of the HCN domain. However, it is also clear that despite the increased resolution of the new Open state structure, the bulk of the HCN domain remains unresolved, suggesting that interaction between this domain and the main body of the channel is weakened in the Open state (See line 236-241).

2. Could the authors explain more about the S4-S5 linker being disordered? Is there any experimental evidence? The data from Yellen's lab about Cd²⁺ crosslinking the missing S4-S5 linker with the C linker in different states seems to contradict the authors' suggestion that the S4-S5 linker is disordered. How would one explain locking the channel in the open state if there is no rigid structure to bind to?

We have added new cryo-EM data to show the densities corresponding to the unwound S4-S5 linker region. Since we now observe density corresponding to this region, it is not floppy, although it loses secondary structure. The residues in the S4-S5 linker are close to the B' helix in the Open state rather than the A' helix (in the Closed state structure). Yellen's Cd²⁺ cross-bridge data were obtained in the SpHCN channel, an ortholog of the mammalian HCN channels and, incidentally, missing the S4-S5 linker region. We must be circumspect about extrapolating those findings to the human HCN channel because there are likely to be significant differences between the structures of the two channels in this region. See the alignment between the two sequences in this region.

3. Figure 4 (and in movie 6 on BioRxiv), the selectivity filter seems to expand a lot in the open state, how come the authors never mentioned it in the text? How does that change the HCN channel's selectivity?

We have now explicitly discussed the conformational changes in the selectivity filter in the text (starting in line 183). We have also carried out three MD simulations lasting 500 ns each to determine the

permeability of these channels to ions. On average, we observe 4-5 permeation events, which is consistent with the low single conductance of these channels. However, these numbers are insufficient to obtain a robust estimate of ion selectivity. Furthermore, there is significant uncertainty regarding the parameterization of various ions, and for historical reasons, potassium ions have been most rigorously parameterized.

4. The narrowest part of the selectivity filter seems to be at C358, which is different from the closed and intermediate states. Any explanations? If this C358 is important for the SF, more experimental data would be needed to validate the open structure. For example, the authors can mutate this C358 and see if it changes ion selectivity.

In the updated model, C358 points away from the central pore and does not form the narrowest part of the selectivity filter region. Mutation of C358 may affect ion selectivity, but it is highly likely that mutations of any of the neighboring residues will also impact ion selectivity. To further validate the structure of the selectivity filter and mechanism of ion selectivity, a combination of experimental and theoretical approaches, presumably involving multiple groups, will be required and is beyond the scope of this manuscript.

As a case point, even though the structure of KcSA was solved more than 25 years ago, the mechanisms of potassium ion selectivity in those channels remain an area of active investigation to this day.

5. In Figure 2C and Line 161, this is a strong statement. The S4 is two helices shorter compared to a previous structure. Is it indeed shorter, or the missing two helices is due to the low-resolution structure?

We have now resolved the missing S4-S5 linker region in the new structure. We now have experimental evidence that S4 and S5 helices become shorter due to the unwinding of the inner end of these helices (see the structure in Figure 5).

6. Fig S7, Since C309 and C385 can form a disulfide bridge in the open state and HCN1-F186C-S264C-C309A produced an intermediate state of the HCN channels, have the authors tested HCN1-F186C-S264C-C385A as well, structurally and functionally?

We did not generate or characterize the HCN1-F186C-S264C-C385A mutant.

7. Also, what is the role of the intermediate state? Any evidence for this intermediate state in other data or physiological function of this state?

The Intermediate state structure was obtained by a mutation of cysteine residue which is involved in disulfide cross-bridge observed exclusively in the Open state. We also show functional data where this mutation destabilizes channel opening consistent with stabilization of the pre-open/intermediate state. Thus, the intermediate state structure is presumably part of the gating pathway during channel activation and deactivation.

Nevertheless, to assuage any lingering concerns, we have addressed it in two ways. First, we refer to this structure as a putative intermediate. Second, we have deemphasized the Intermediate state throughout the manuscript. The structures were initially annotated based on the structural features as is the standard practice in the field. For the Open-state structure, we could carry out further validation by running MD simulations to observe permeation events. However, this is not possible for the Closed or Intermediate state structures. To definitively establish whether such a structure truly corresponds to an Intermediate state would be technically challenging, if not impossible, with current methods.

8. The authors keep mentioning that the S4-S5 inner ends keep the channels closed, please elaborate more.

We have clarified this in line 334-339. Briefly, we propose that the loss of secondary structure in intracellular ends of S4 and S5 helices in the Open state creates space for S6 to undergo iris-like displacement resulting in channel opening.

Reviewer #2 (Remarks to the Author):

More specific points:

1) Line 102. The transmembrane helices S5 and S6 are not well-resolved. There is almost no density for side chains in these helices, which is quite poor for the nominal 3.73Å resolution.

As suggested by the reviewers and the editors, we sought to improve the resolution of the open state map. Orientation bias severely limited the 3D reconstruction of the open-state map. Due to a majority of the particles being top views, the density in the open state was highly anisotropic, and some interpretations relied on the deepEMhancer-sharpened map. By collecting additional cryo-EM data using tilt series data collection, we were able to greatly improve orientational sampling, which in turn improved classification and resulted in a higher-quality final reconstruction with a global resolution of 3.6 Å. This map enabled less ambiguous interpretation without the use of deepEMhancer.

Viewing direction distribution plots reveals that orientational bias in the final map is significantly reduced by the addition of tilt series data prior to 3D classification. In particular, the map has been drastically improved between the TMD and CNBD.

Above, we show the model vs map for S5 and S6, which reveal critical features for assigning registry. The resolution in S6 is sufficient for placing sidechains, while bulky sidechains in S5 and disulfide bond with S6 enable clear placement.

2) Line 105. By looking at cryo-EM density, it is hard to conclude that the registry in S4 is unambiguous. Instead, it is rather ambiguous. The N-terminal part of S4 is asking for being moved up by 1.5-2 helical turns into unaccounted density above. The density bump assigned to R270 can easily correspond to W221 or R273. The entire C-terminal part of S4 fits very poorly into the map, emphasized by unaccounted density next to I278. With such poor overall placement, it is very hard to justify a bump of density-based assignment of W281 as well.

We appreciate this point, as the assignment of S4 in the previous map was difficult and complicated by the lack of sidechain density. In the improved open state map, the C-terminal part of S4 can be assigned with high certainty. There is no longer unaccounted-for density at the top of S4, while the upper half of S4 has shifted up by about 1 helical turn. The previous map likely had extra unaccounted-for density at the top of S4 due to z-axis streaking due to the preponderance of top views in the data.

3) Line 107. S3 does not look like a single helix. It appears to be two consecutive helices tilted relative to each other. Alternatively, S3 can be called a single helix severely kinked in the middle.
4) Lines 116-122. When deciding between what was more important for the open state, detergent-lipid mixture or metal cross-bridge, the authors solved the structure in reducing conditions but in the same detergent-lipid mixture. However, to prove their point that the oxidizing conditions primarily facilitate the

open state, the authors need to solve the structure in the presence of Hg²⁺ and DDM and show that the channel is still open. Otherwise, both factors are important contributors to their open channel conformation.

Agreed. We have now clarified that we are assessing the role of the metal cross-bridge in stabilizing the open state. Based on our structure, we can state that a metal cross-bridge is necessary to obtain the open-state structure but is not a sufficient condition. Indeed, the same metal cross-bridge did not lead to an Open state structure in digitonin, as reported elsewhere.

5) Lines 132-134. S5 and S6 are shown together in Extended Data Fig. 2, where one can appreciate the density for the putative disulfide bridge. In Extended Data Fig. 5, S5 and S6 are shown separately and there is no chance to make an opposite observation. Please include a comparable view of S5 and S6 in Extended Data Fig. 5.

We have generated a comparable view of S5 and S6 for the closed density cutouts and it is shown below as well as incorporated into Extended Data Fig. 5

6) Lines 161-162. If S4 moves up in the cryo-EM density (see comment 2 above), would it really be shortened?

See our response to point #2). The new densities clearly show that S4 helix is unwound at the C-terminal end and consequently, is shorter in the Open state compared to the Closed state.

7) Lines 187-188. Here, Y386, V390, and Q398 are called gate residues. Should these be V390, T394, and Q398 instead (see Figure 4)? The authors also mention the rotation of these residues. However, how do they know that these residues rotate, if the side chains are not visible in the cryo-EM density (see, for example, V390)?

While Y386, V390, and Q398 represent the narrowest constrictions in the open state, they are not the gate residues, and we have adjusted our wording to reflect this. Regarding rotation – Y386 is unambiguously placeable in the map (see our response to comment 1), and V390 and Q398 are deflected in response to the iris-like displacement of the backbone of S6. We have adjusted our wording to indicate that the

residues are displaced by the rotational expansion of the S6 helices rather than rotating individually at the sidechain level (see Lines 173-175).

8) Line 204. Reference 32 reports crystal, not cryo-EM structures.

Good catch. Sorry about that. We have removed that reference. We were referring to cAMP-bound structures of the full-length channel.

9) Line 205. To better illustrate that the CNBD structure in all three states is identical, please add a panel with their superposition to Figure 5.

We have clarified that the structures are similar but not identical (Line 227). There are key differences between the CNBDs of Open and Closed state structures. These are highlighted in Fig. 4c and d.

10) Lines 233 and 393. The PDB code does not match the reference.

Thank you for catching that. We now have the correct PDB code.

11) Lines 250-255. Based on the resolution, model building in this region is very approximate. Accordingly, it seems to be a stretch to discuss how much the disulfide bond is strained guided by such a low-resolution model.

The exact energy and strain of the disulfide bond between S5-S6 is highly dependent on fit, and given that the local resolution is insufficient to pin down the specific geometry of the bond, we agree that absolute quantitative estimates based on our fit are of limited utility. We can say that the bond is a disulfide staple ($\chi_2 > 0$, $\chi_3 < 0$, $\chi_2' > 0$), and that it is strained in our current best-fit model. Therefore, we have refrained from attempting energetic calculations based on our model (see line 300-303).

12) Figure 1 and further. The colors chosen for the closed and open conformations are very similar. It would be beneficial to change the color of one (to blue?) to make comparisons clearer.

We agree and we have updated the colors. See new Figure 1.

13) Please label structures in Extended Data Figure 3 and/or indicate which one is which in the figure legend.

The structures are labeled in the Figure. It is now Supplementary Figure 10.

REVIEWERS' COMMENTS

Reviewer #1 (Remarks to the Author):

The authors have done a good job responding to my comments. However, there are some unclarities in figures and legends that needs to be resolved. Also one control experiment needs to be done on a more relevant hHCN1 mutant.

1. what is pink line in Figure 3b?
2. something is missing in Fig 4 legend c and d. and what is in gold color?
3. Also, what does the thin arrow between B' and HCN mean?
4. Also, 4C is not described, bottom view or top view?
5. Fig. 6. Where is C linker that you talk so much about in the Discussion? Or is it not important?
6. Also, is it the bend in S4 that is important for opening or the unwinding of the S4 and S5 helices? Seems to me that the bending is more important looking at your movies...
7. Petrol as color name is very unusual, please change to turquoise, or something more common.
8. Suppl. Fig. S6. You should compare hHCN1-EM F186C-S246C with hHCN1-EM F186C-S246C-C309A to really show the effect of the C309-C385 disulfide bond. Now you compare with hHCN1-EM only....
9. Suppl. Fig. 14. Something wrong in a left...is it really depolarized open? not closed?
10. Also the PDB name links to the intermediate state in BioRx

Reviewer #2 (Remarks to the Author):

The authors have addressed my original comments and I have no further comments.

Reviewer #3 (Remarks to the Author):

The manuscript presents the structure of HCN channels in the open conformation. This work enables novel insights into the mechanisms of gating and ion permeation in these channels. Cryo-EM and Molecular dynamics simulation experiments are well done and present interesting and novel findings that will advance the field.

The paper would benefit from more detailed analysis of ion permeation/conductance. (eg. are ions offset from the center of the pore, number of waters co-ordinating permeating ions, etc.). More importantly, comparisons to other work in this area should be presented, particularly since various similarities and differences exist. For example, the authors present the carbonyl groups of I359 and C358 as co-ordinating sites for potassium ions, but do not reference previous similar findings by other groups. Additionally, PMF and other calculations from closed-state studies suggest favorable binding at C358, whereas here the authors present a small energy barrier at this residue.

J Gen Physiol. 2023 Oct 2;155(10):e202313364; functon(Oxf). 2022 Apr 22;3(3):zqac019; Biophys J. 2022 Jun 7;121(11):2206-2218.

2. In reference to the authors' responses to previous reviewers: The role of the Cys in the selectivity filter of HCN4 and HCN2 channels were previously explored and demonstrated to play a major role in ion selectivity.

PLoS One. 2009 Nov 5;4(11):e7712; Sci Rep. 2012;2:894; Biophys J. 2022 Jun 7;121(11):2206-2218.

Response to reviewers:

We thank all three reviewers for their comments and feedback. See our responses below.

Reviewer #1 (Remarks to the Author):

1. what is pink line in Figure 3b?

The pink line in Fig 3b refers to Y386. We have labeled it in the figure and updated the Figure caption.

2. something is missing in Fig 4 legend c and d. and what is in gold color?

The gold/tan color in Figure 4 indicates the HCNa helix, which is resolved in the closed state but not in the open state. We noticed that the figure legend did not accurately capture this detail, so we have updated it accordingly.

Also, what does the thin arrow between B' and HCN mean?

This was an error. The arrow mirrors the arrows showing HCN domain displacement as seen in the other three subunits. We have fixed this in revised Figure 4.

4. Also, 4C is not described, bottom view or top view?

Figure 4c is a top view, and we have updated the figure legend to indicate this.

5. Fig. 6. Where is C linker that you talk so much about in the Discussion? Or is it not important?

Our goal in Fig 6 is to highlight how the repositioning of the transmembrane helices driving channel gating. The C-linker is important but not essential. Also, for the sake of clarity, it is better to show the transmembrane helices without the C-linker especially the views from the bottom.

6. Also, is it the bend in S4 that is important for opening or the unwinding of the S4 and S5 helices? Seems to me that the bending is more important looking at your movies...

Based on the structures, we cannot say one way or the other whether the unwinding or bending is more important. We observe both conformational transitions in our Open-state structure and have not explicitly tested which structural change has a greater impact on channel gating. New experiments will be needed to sort this out in the future.

7. Petrol as color name is very unusual, please change to turquoise, or something more common.

The color name in the figure legends (Fig. 6 and Suppl. Fig. 14) have been updated to turquoise.

8. Suppl. Fig. S6. You should compare hHCN1-EM F186C-S246C with hHCN1-EM F186C-S246C-C309A to really show the effect of the C309-C385 disulfide bond. Now you compare with hHCN1-EM only...

For readability, we will use the following abbreviations:

WT (for HCN1-EM), CC (hHCN1-EM F186C-S264C) and CCA (hHCN1-EM F186C-S264C-C309A)

We do not believe this additional experiment proposed by the reviewer is needed for two reasons.

- A. The difference in voltage of half-maximal activation ($V_{0.5}$) between the WT and CCA mutant is only 8 mV, which is not a substantial difference. The MacKinnon lab characterized the CC mutant and found that the $V_{0.5}$ is shifted by 4 mV towards more negative potentials compared to hHCN1-EM (WT) (Lee & MacKinnon 2019, DOI: 10.1016/j.cell.2019.11.006). Since we used *Xenopus laevis* oocytes as an expression system as opposed to HEK in the other study, we cannot compare them directly. However, if both mutants are normalized to the corresponding recordings of the WT, we find that the CCA mutant shows a 4 mV leftward shift. However, these shifts are too small to be interpreted meaningfully.
- B. We have already stated in Pg 5 line 15, these measurements do not allow us to estimate the energetic impact of the loss of the disulfide bond on the channel opening. *“To obtain a complete estimate of the energetic cost of disulfide cross-bridge in stabilizing the Open state, an assessment of the open channel probability (P_o) is required, which is technically challenging given the low single-channel conductance²² of these channels.”*

We have further toned down the significance of the disulfide by making the following changes:

We replaced the sentence on page 11 (line 21) *“Our functional data also suggests that the disulfide bond may be important for stability of the open pore.”* and therefore replaced it with *“Our functional data suggest that the disulfide bond marginally stabilizes the open pore but is not critical.”* A new sentence in pg. 11 (line 25): *“Nevertheless, we cannot rule out the possibility that the disulfide bond observed in our structure is due to our experimental conditions and does not have a physiological role.”*

9. Suppl. Fig. 14. Something wrong in a left...is it really depolarized open? not closed?

Thanks for catching this error. Indeed, the left panel depicts the closed state. We corrected Supplementary Figure 14 correspondingly.

10. Also the PDB name links to the intermediate state in BioRx

We have thoroughly re-checked all PDB names for accuracy. All structures obtained in this study were registered with the PDB when the preprint on BioRxiv was released. The improved maps and structures were re-evaluated (Supplementary Table 1) and resubmitted to the PDB, retaining their original PDB IDs.

Reviewer #2

The authors have addressed my original comments and I have no further comments.

Thank you!

Reviewer #3

1. The paper would benefit from more detailed analysis of ion permeation/conductance. (eg. are ions offset from the center of the pore, number of waters co-ordinating permeating ions, etc.).

We thank the reviewer for pointing this out. We have added two new panels in Suppl Fig 12 showing ion offset and water coordination numbers. The Results section of the main manuscript was also updated (see Pgs.6-8).

2. More importantly, comparisons to other work in this area should be presented, particularly since various similarities and differences exist. For example, the authors present the carbonyl groups of I359 and C358 as co-ordinating sites for potassium ions, but do not reference previous similar findings by other groups. Additionally, PMF and other calculations from closed-state studies suggest favorable binding at C358, whereas here the authors present a small energy barrier at this residue.

J Gen Physiol. 2023 Oct 2;155(10):e202313364; Funct (Oxf). 2022 Apr 22;3(3):zqac019; Biophys J. 2022 Jun 7;121(11):2206-2218.

We agree! We have now cited all the above references in Pg. 8 where we describe the coordination by I359 and C358.

3. In reference to the authors' responses to previous reviewers: The role of the Cys in the selectivity filter of HCN4 and HCN2 channels were previously explored and demonstrated to play a major role in ion selectivity.

PLoS One. 2009 Nov 5;4(11):e7712; Sci Rep. 2012;2:894; Biophys J. 2022 Jun 7;121(11):2206-2218.

Thanks for pointing that out.